# DynamicEval: Rethinking Evaluation for Dynamic Text-to-Video Synthesis

## Abstract

Existing text-to-video (T2V) evaluation benchmarks, such as VBench and Eval-Crafter, suffer from two main limitations. (i) While the emphasis is on subject-centric prompts or static camera scenes, camera motion which is essential for producing cinematic shots and the behavior of existing metrics under dynamic motion are largely unexplored. (ii) These benchmarks typically aggregate video-level scores into a single model-level score for ranking generative models. Such aggregation, however, overlook video-level evaluation, which is vital to selecting the better video among the candidate videos generated for a given prompt. To address these gaps, we introduce DynamicEval, a benchmark consisting of systematically curated prompts emphasizing dynamic camera motion, paired with 45k human annotations on video pairs from 3k videos generated by ten T2V models. DynamicEval evaluates two key dimensions of video quality: background scene consistency and foreground object consistency. For background scene consistency, we obtain the interpretable error maps based on the Vbench motion smoothness metric. Our key observation based on the error maps is that while the Vbench motion smoothness metric shows promising alignment with human judgments, it fails in two cases, namely, occlusions/disocclusions arising from camera and foreground object movements. Building on this, we propose a new background consistency metric that leverages object error maps to correct two major failure cases in a principled manner. Our second innovation is the introduction of a foreground consistency metric that tracks points and their neighbors within each object instance to better assess object fidelity. Extensive experiments demonstrate that our proposed metrics achieve stronger correlations with human preferences at both the video level and the model level (an improvement of more than 2% points), establishing DynamicEval as a more comprehensive benchmark for evaluating T2V models under dynamic camera motion.

## 1 Introduction

The rapid advancement of foundational text-to-video models (Zheng et al., 2024; Yang et al., 2024; Kong et al., 2024; Wan et al., 2025) has necessitated the development of automatic evaluation metrics that correlate highly with human preferences. However, despite the significant developments in the space of video models, the development of automatic metrics has severely lagged. Recent works such as VBench (Huang et al., 2024a) and EvalCrafter (Liu et al., 2024b) introduced evaluation prompt suites along with automatic metrics that assess several dimensions of video quality, including background consistency, object consistency, text alignment, and color. While these benchmarks provide broad coverage, their prompts are predominantly generic and subject-centric, overlooking the role of camera motion. Another critical limitation of current evaluation practices is their exclusive focus on model-level assessment, where average win ratios are computed for each model, both in human evaluation and automatic metrics. Metrics are typically evaluated based on how well their model rankings correlate with average human preference rankings. While such model-level analyses may yield high human correlation scores, they fail to capture the actual alignment between automatic metrics and human preferences at the individual video level. Video-level assessment can enhance the effective T2V generation quality either by selecting high-quality videos from those generated for a given prompt, or by optimizing models using video-level metrics as reward signals.

To address the lack of systematic evaluation for dynamic scenes in text-to-video (T2V) generation, we introduce DynamicEval, a comprehensive benchmark designed to assess video generation quality under dynamic camera motion. DynamicEval consists of two key components: (1) a procedurally generated prompt suite featuring highly detailed descriptions that explicitly specify camera motion, and (2) 45k high-quality human annotations across 3k videos generated by ten T2V models using this prompt suite. We conduct a large-scale subjective study in which human annotators compare pairs of videos generated from the same prompt, evaluating them across multiple quality dimensions. Throughout this paper, we refer to videos that exhibit explicit camera motion as dynamic scenes or dynamic videos. For evaluation, we specifically focus on two critical dimensions of dynamic video quality: (1) background (BG) consistency and (2) foreground (FG) object consistency. Existing metrics that evaluate these two dimensions, in particular, VBench background consistency and subject consistency (Huang et al., 2024a), compute feature similarities across consecutive frames using pretrained deep networks. These methods offer limited fine-grained spatial awareness and long-term temporal context, as they rely on global features and pairwise similarities, respectively. For instance, the background consistency metric computes frame-wise CLIP (Radford et al., 2021) score similarities, but the global nature of CLIP features limits their ability to capture fine-grained temporal inconsistencies in the background at a pixel-level. Similarly, the subject consistency metric computes similarities of DINO (Caron et al., 2021) features between consecutive frames, limited by low-resolution attention maps relative to frame sizes, which reduces their fine-grained spatial awareness. To overcome these challenges, we propose fine-grained metrics using pixel-level, interpretable tools for improved spatial detail and temporal consistency.

For background (BG) consistency, we first investigate the common evaluation metrics, in particular, VBench motion smoothness (VB-MS) (Huang et al., 2024a), which relies on the RAFT optical flow model Teed & Deng (2020). Our analysis reveals that, despite its simplicity, VB-MS shows promising alignment with human preference, while also providing a pixel-level quality map for evaluation. However, this metric accounts for the entire frame including foreground objects, and yields large errors near occlusions and disocclusions caused by camera motion. We overcome these limitations by debiasing motion smoothness through isolating the foreground objects and removing occlusion-related background pixels, ensuring a temporally stable consistency measure independent of moving objects and errors emerging from camera motion. For foreground (FG) object consistency, an ideal metric should isolate objects, remain robust to camera/object motion, and capture long-term object details. The VBench subject consistency metric, leveraging DINO feature similarity across frames, fails to capture these nuances. We propose a fundamentally different approach to measure FG object consistency by tracking multiple points on the foreground objects using CoTracker (Karaev et al., 2024) and monitoring their nearest neighbors. Our subject consistency metric is then defined by analyzing the smoothness of distances between these tracked points over time. This method is highly effective in capturing subtle deformations of the object throughout the video, ensuring long-term temporal understanding. Finally, using our DynamicEval benchmark, we demonstrate that our proposed metrics achieve a significantly higher agreement with human evaluations compared to existing metrics on both video-level and model-level evaluation. Our key contributions are as follows:

- We propose a comprehensive human evaluation suite, DynamicEval, comprising 100 procedurally curated prompts with diverse camera motions, along with 45k human annotations on 3k videos generated by ten T2V models, designed for video-level evaluation.
- In contrast to existing baselines that solely rely on deep feature based metrics, which fail to capture fine-grained spatial awareness and long-term temporal context, we propose methods that provide pixel-level evaluation. For BG scene consistency, we mitigate the two key factors (occlusions/disocclusions and FG objects) that bias motion smoothness in dynamic videos. For FG object consistency, we evaluate the temporal smoothness of neighboring tracks within an object, enabling robustness to camera and object motion.
- We conduct extensive experiments on DynamicEval dataset to demonstrate that our proposed metrics achieve stronger agreement with human preferences than baseline metrics (an improvement of more than 2% points), across both video-level and model-level evaluations. In addition, our large-scale video-level annotations on dynamic videos can serve as a valuable resource for developing new metrics and advancing T2V generation.

## 2 RELATED WORKS

**Text-to-Video Generative Models.** In the last few years, the field of video generation has experienced a great impetus with diffusion-based generative models (Blattmann et al., 2023; Xing et al., 2024; Chen et al., 2023; Brooks et al., 2024; Bar-Tal et al., 2024; Polyak et al., 2024; Yang et al., 2024; HaCohen et al., 2024) to generate realistic videos based on textual conditions. In particular, following the seminal works of Brooks et al. (2024); Gupta et al. (2024), there has been large developments in Diffusion Transformer (DiT) (Peebles & Xie, 2023) based video foundation models both open-source (Zheng et al., 2024; Lin et al., 2024; Yang et al., 2024; Kong et al., 2024; Wan et al., 2025; HaCohen et al., 2024) and commercial (RunwayML, 2024b; LumaLabs, 2024; Hailuo, 2024; Adobe, 2025; Deepmind, 2025) variants, that can generate long and high-resolution videos. Considering the rapid development and commercialization in this field, it becomes extremely important to develop evaluation criteria and metrics to judge the generation quality of video foundation models.

**Benchmarks and Datasets.** The availability of the text-to-video models has led to the development of large-scale well-curated evaluation benchmarks studies like VBench (Huang et al., 2024a), VBench++ (Huang et al., 2024b), EvalCrafter (Liu et al., 2024b), DEVIL (Szeto & Corso, 2022) and GenAIarena (Jiang et al., 2024). VBench, EvalCrafter and DEVIL provide a large suite for model-level evaluation metrics across several dimensions, with prompts that produce predominantly subject centric and static scenes. Model-level evaluations suffer from aggregation of preferences across all prompts (or videos) in the suite. Different from these benchmark studies and suites, DynamicEval provides a comprehensive suite of pairwise video comparisons that can be used to assess automated metrics with human preferences specifically for dynamic scenes.

**Evaluation Metrics.** For dynamic scenes, we focus on the two key dimensions of video quality: background scene consistency and foreground object consistency. Vbench (Huang et al., 2024a) provides a background consistency metric based on the similarity scores between CLIP (Radford et al., 2021) embeddings of consecutive frames. While it captures the overall content consistency across frames, it fails to detect localized background distortions that require pixel-level analysis. MEt3R (Asim et al., 2025) introduces a metric based on the computation of 3D point clouds for consecutive frames. Although this metric operates at a fine-grained level, it is vulnerable to errors from inaccurate 3D point estimation by DUSt3R (Wang et al., 2024) on generated frames. Vbench also introduces a subject consistency metric that relies on DINO (Caron et al., 2021) feature similarities across consecutive frames. DINO models are self-supervised transformer models that are found to attend more to the primary objects in a frame. However, the reduced resolution of attention maps compared to the original frame size limits their ability to capture fine-grained object details. Additionally, since they are computed independently at the frame level without leveraging neighboring frames, they are highly sensitive to scene variations, particularly in dynamic videos. Motivated by these limitations, we propose pixel-level and temporal tracking based methods that move beyond feature-level approaches, enabling fine-grained and interpretable evaluation.

## 3 DYNAMICEVAL: DATASET

Existing generated video evaluation benchmarks (Huang et al., 2024a; Liu et al., 2024b) introduce prompts that are often subject-centric and depict relatively static scenes with little-to-no camera motion. This highlights the need for a prompt suite that targets T2V generation of scenes involving significant camera motion. For the purpose of this work, we use the term *'dynamic scenes/videos'* to refer to videos with substantial camera motion, unless specified otherwise. We introduce DynamicEval, a fine-grained generated video evaluation dataset that focuses on evaluating dynamic videos by carefully curating text prompts that describe various camera motions in different scenes and subject descriptions. Our dataset includes pairwise video preference annotations along two key dimensions of dynamic video quality: (1) background scene consistency and (2) foreground object consistency. We describe our dataset construction in detail in the subsequent sections.

### 3.1 PROMPT CURATION

To generate diverse scenes with camera motion, we introduce a procedural prompt curation strategy that incorporates camera motion. Following existing works (Lee et al., 2023; Huang et al., 2024a; Bakr et al., 2023), we collect various keywords for scene elements by randomly sampling across

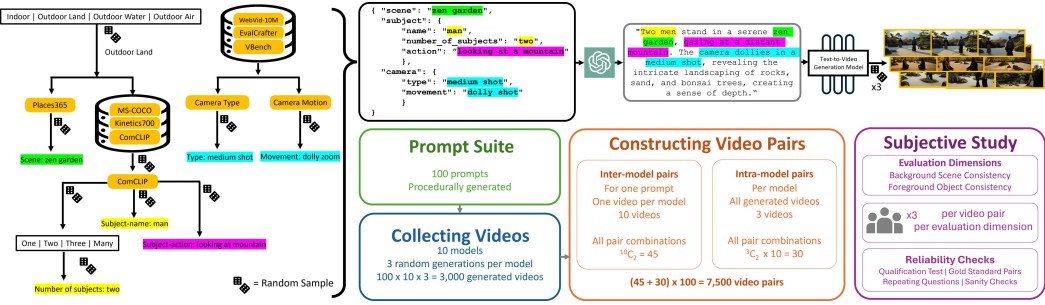

Figure 1: Prompt curation: Scene elements from databases (orange) are sampled into a metadata (JSON format), which GPT-4o converts into descriptive prompts. Dataset: Video pairs generated from a common prompt are annotated via a subjective study.

three key aspects of dynamic videos: (i) background scene, (ii) primary object(s), and (iii) camera movement, as illustrated in Fig. 1. The camera attributes specify a camera type and motion, which we collect from camera-related keywords extracted from prompt benchmark datasets (Bain et al., 2021; Liu et al., 2024b; Huang et al., 2024a) (see supplementary for details on collection of keywords for each aspect). We randomly sample keywords and their paired attributes to construct scene metadata in JSON format and prompt GPT-4o (Hurst et al., 2024) to generate descriptive prompts from the metadata. This approach procedurally generates complex scenes that feature highly diverse background settings, object types, and their motions, as well as camera movements. We construct our benchmark prompt suite of 100 prompts using this pipeline as illustrated in Figure 1.

## 3.2 SUBJECTIVE STUDY

We generate videos using our prompt suite with ten latest state-of-the-art T2V models which includes both open-source (OpenSora (Zheng et al., 2024), OpenSoraPlan (Lin et al., 2024), CogVideoX (Yang et al., 2024), HunyuanVideo (Kong et al., 2024), Wan2.1 (Wan et al., 2025), and LTXVideo (HaCohen et al., 2024)) and closed-source (DreamMachine (LumaLabs, 2024), Pika (PikaLabs, 2025), Runway Gen2 (RunwayML, 2024a), and Runway Gen3-Alpha (RunwayML, 2024b)) models. From open-source models, we ensure high quality generation by selecting the largest-parameter variants of each model. We collect three videos per prompt per model, totaling 3k videos, making this a comprehensive T2V dataset with both inter-model and intra-model video-level quality comparisons. For human preference on different quality aspects of dynamic videos, we conduct a large-scale subjective study on generated video pairs using Amazon Mechanical Turk (AMT). We study two key dimensions in the generated videos: background scene consistency and foreground object/subject consistency.

**Background scene consistency.** In generated videos with significant camera movement, the background may undergo unnatural morphing or stretching, leading to localized distortions. We show an example of a generated video demonstrating low background scene consistency in Fig. 3.

**Foreground object/subject consistency.** Foreground objects in generated videos may exhibit unnatural shape changes across frames, even if the background scene remains consistent. This evaluation dimension captures how foreground objects/subjects remain consistent throughout the scene. For example, under camera motion, a generative model may fail to preserve the consistency of a human face, as illustrated in the second example of Fig 5.

**Crowd-sourced Human Subjective Study.** We conduct a subjective study where we present pairs of videos generated from the same prompt and ask participants to select the preferred video for each evaluation dimension. With 3 generations each from ten different T2V models, we obtain 30 videos per prompt. Instead of exhaustively collecting all $^{30}C_2$ pairs, we sample 45 inter-model and 30 intra-model pairs. (details in supplementary material), resulting in 7.5k video-pair comparisons per evaluation dimension. To ensure reliable fine-grained annotations we follow standard practices (Sinno & Bovik, 2019; Hosu et al., 2017) and employ multiple reliability checks, including an initial qualification study, gold standard pairs, repeated questions, and content-related questions. We collect three human annotations per comparison, yielding 45k fine-grained human annotations.

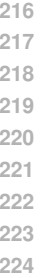
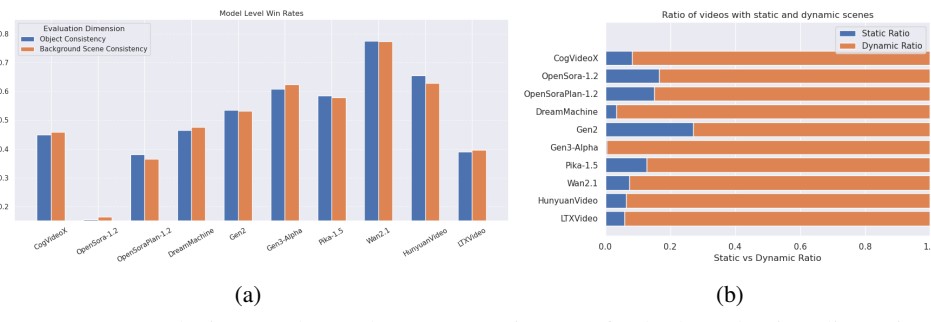

(a)                                   (b)

Figure 2: Dataset analysis: (a) shows the average win rates for both evaluation dimensions. (b) illustrates the percentage of video samples in each model that are static and dynamic.

## 3.3 DATASET ANALYSIS

**Model-level human preference.** We analyze the average model-level win ratios of videos across inter-model pairs in Fig. 2a. The win ratio calculates the fraction of times a video is selected out of all its comparisons with other videos, with a higher win ratio indicating better model performance. We find that older open-source models generally achieve lower win ratios on both evaluation dimensions, whereas closed-source models generate higher quality videos with better background and foreground object consistency. Notably, the latest open-source model, Wan2.1 (Wan et al., 2025) outperforms all other models across both dimensions.

**Presence of static scenes in the video generations.** Despite explicitly providing prompts with camera motion descriptions, some models fail to generate dynamic videos. To identify static and dynamic scenes, we average the variance of point tracks from CoTracker (Karaev et al., 2024) across frames to obtain a camera-motion metric. Through empirical analysis, we find that the bottom $10\%$ of videos by camera motion values corresponds to static scenes with very little to no camera motion. We treat these as static videos and the remainder as dynamic. The ratio of static-vs-dynamic videos per model is shown in Fig. 2b. We find that most open-source models and an older closed-source model (e.g., Runway Gen2) generate many static scenes despite camera motion explicitly mentioned in the prompt. We retain static videos in the evaluation to ensure our metrics perform well in both static and dynamic scenarios.

## 4 DYNAMICEVAL: METRICS

We introduce two metrics on the key dimensions of dynamic video quality. For background scene consistency, we leverage dense optical flow based measures that capture finer frame-level distortions in the background scene. In contrast, for foreground object consistency, the metric needs to keep track of object shape deformations across time. We isolate objects in pixel space and employ point tracking methods for evaluation.

## 4.1 BACKGROUND (BG) CONSISTENCY

In this section, we first review the commonly used evaluation metrics introduced in VBench (Huang et al., 2024a), EvalCrafter (Liu et al., 2024b), and Met3R (Asim et al., 2025). These metrics have been shown to perform well on existing prompt suites, which are predominantly subject-oriented and involve very little (or static) camera motion. To assess their limitations, we evaluate them on our database to analyze how they behave when applied to videos with significant camera motion.

**Background consistency metrics.** We evaluate the baseline metrics on the background scene consistency dimension of our DynamicEval dataset by computing the pairwise video preference of each method with respect to subjective human preferences. From VBench (Huang et al., 2024a), we eval-

Table 1: Pairwise video selection accuracy of existing metrics on BG scene consistency.

| Metrics | VB-BG | VB-flickering | VB-MS | EC-semantic | Met3R |
|---|---|---|---|---|---|
| Accuracy | 56.0 | 51.3 | 53.7 | 52.0 | 54.9 |

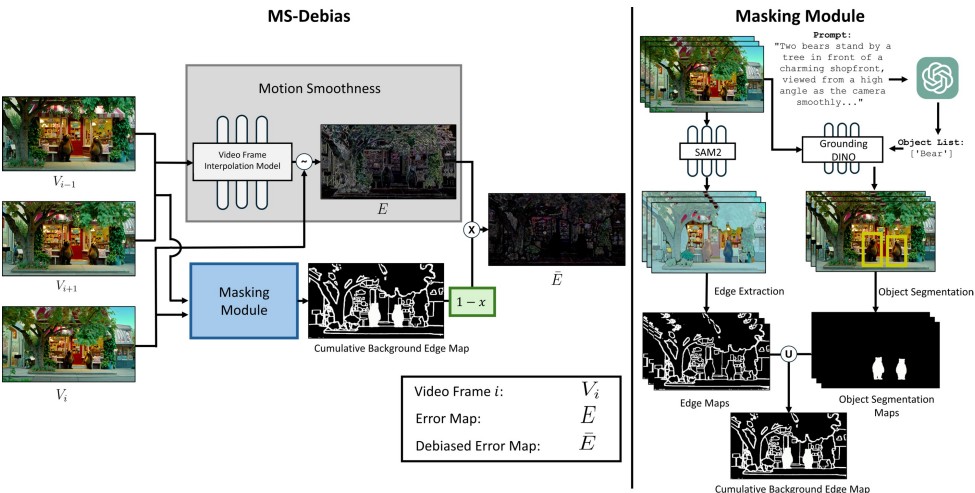

Figure 3: Motion Smoothness error maps: The zoomed in regions show localized distortions visible across frames. VB-MS shows large errors near edges and foreground objects, suppressing the localized distortions. After debiasing, the localized distortions are visible.

Figure 4: MS-Debias obtains debiased motion smoothness error maps by masking out foreground objects and occlusions. It is applied at multiple scales with Gaussian pyramid downsampling.

uate background consistency (VB-BG), flickering (VB-flickering), and motion smoothness (VB-MS), and from EvalCrafter (Liu et al., 2024b), we include the semantic consistency (EC-semantic) metric. In addition, we also evaluate Met3R (Asim et al., 2025). Among the different metrics provided by VBench and EvalCrafter, we selected those that can serve as representative proxies for background scene consistency. The evaluation results presented in Table 1shows that VB-BG, VB-MS, and Met3R perform the best of the evaluated metrics. Due to its simplicity and access to pixel-level evaluation, we further analyze VB-MS for background consistency. VB-MS leverages motion priors from a video interpolation model to assess how naturally pixels move in a scene.

**Analysis of VB-MS.** As VB-MS is a pixel level metric, it captures localized distortions in the background very well. Motion smoothness uses the optical flow model, RAFT Teed & Deng (2020), to predict an intermediate frame between two alternate frames. The absolute difference between the predicted frame and the original frame provides an error map, which is spatially and temporally averaged to obtain the inconsistency score. To understand how the motion smoothness captures localized distortions, we analyze the error map of a generated video in Figure 3. The error maps reveal local inconsistencies in the background scene. We also observe that although the error map captures localized issues, it is highly influenced by regions near object edges. This behavior primarily arises due to object occlusions with moving camera. Optical flow reconstruction often fails near occlusions under camera motion, which increases the motion smoothness error. We identify object edges to be a major contributing factor biasing the motion smoothness error whenever camera motion is present. Further, the foreground objects also significantly contribute to the error, as object motion is harder to predict with optical flow, and does not reflect background consistency. We leverage these observations and discuss techniques to debias and improve motion smoothness by carefully controlling the contributions of these errors near the occluded edges and foreground objects.

**Debiasing motion smoothness metric.** To address the camera motion and foreground object bias in motion smoothness, we carefully construct masks around object edges (to account for bias near

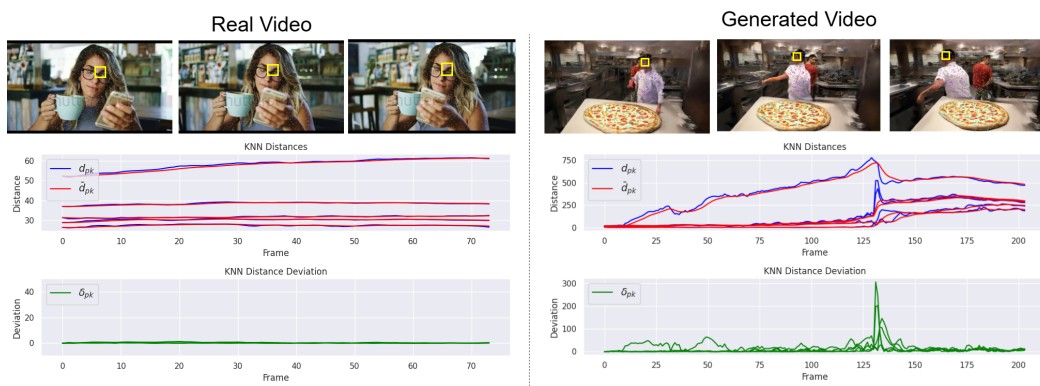

Figure 5: Deviation of neighbor tracks on real video vs generated video with object distortions. Blue plot: Distance between a nearest neighbor and a candidate point. Red plot: Moving average of blue. Green plot: the deviation between blue and red.

occlusions or dis-occlusions) and foreground object masks and reduce their contribution to the error computation as shown in Figure 4. To detect object boundaries, we employ the auto-object detector of SAM-2 (**?**) and propagate the object segmentation masks across all the frames. The segmentation masks are then converted into edge boundaries by applying a morphological gradient, and thickened by dilation, to obtain the final edge map, $M_i^{\text{edg}}$, where $i \in \{1, 2, \cdots, F\}$ refers to the frame number. To detect foreground objects, we leverage the prompt that was used to generate the video. We pass the prompt to an LLM (ChatGPT-4o) to extract object mentions and classify them as static or dynamic. We pass the list of moving object names into the GroundingDINO (Liu et al., 2024a) model to localize these objects in the video. The localizations are further transformed into masks and propagated across all frames using SAM2 (**?**) to obtain the video segmentation mask for each object as $M_{in}^{\text{obj}}$; $n \in \{1, 2, \cdots, N\}$, where $N$ denotes the number of objects in the scene. The individual object masks are merged into a final object mask, $M_i^{\text{obj}}$,

$$M_i^{\text{obj}} = \cup_{n=1}^{N} M_{in}^{\text{obj}} \tag{1}$$

The final debiased error map, $\bar{E}_i$, is computed as,

$$\bar{E}_i = E_i \times \left( 1 - \left( M_i^{\text{edg}} \cup M_i^{\text{obj}} \right) \right) \tag{2}$$

where $E_i$ is the error map obtained from VB-MS for the $i^{\text{th}}$ frame. We note that our debiased error maps are devoid of the localized issues as shown in Figures 3. Thus, the debiased error map can be used as a tool to evaluate generated videos at a pixel level. Additionally, to mimic the multi-scale processing capabilities of the human visual systems (Wang et al., 2003; Soundararajan & Bovik, 2012; Li et al., 2016), we apply our debiased motion smoothness to multiple downscaled versions of videos and aggregate the scores to obtain the final metric.

## 4.2 FOREGROUND (SUBJECT/OBJECT) CONSISTENCY

In our analysis, we think that the only existing metric that actually focuses on the foreground consistency is the VBench subject consistency (VB-SC) metric (Huang et al., 2024a). VB-SC computes the similarities of the DINO (Caron et al., 2021) characteristics in consecutive frames. These features are computed at the frame-level and rely on attention maps that are much smaller than the frame sizes, making them ill-suited for evaluating the temporal aspects of subject consistency in finer detail. To address this, instead of using the features from pretrained model (like DINO), we use point tracks, which capture fine-grained details and long-term temporal context. In Figure 5 we observe low neighbor track deviation in real videos, while with inconsistent objects in generated video, the deviation is high. Motivated by this, we propose Track-FG as shown in Figure 6. We leverage the object masks $M_{in}^{\text{obj}}$ used earlier in Section 4.1. Here, $i$ corresponds to the frame number and $n$ corresponds to the object index. We use a state-of-the-art point tracking model, CoTracker (Karaev et al., 2024), to track randomly sampled points inside the object masks. While designing the evaluation metric using these tracks, we ensure two major requirements. First, the metric should evaluate the consistency of neighboring tracks; second, it should remain invariant to global object motion and camera

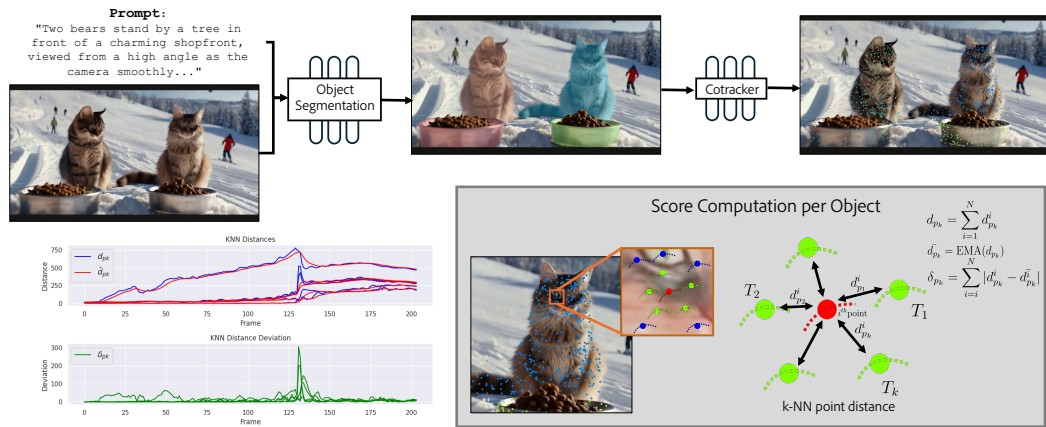

Figure 6: Object Tracking Score framework: We track the points inside an object mask across the video. TrackerFG is obtained by computing the distances between neighboring tracking points.

movement. For $N$ objects in the video, let all the point tracks be denoted as $T_p^n; p \in \{1, 2, \cdots, P_n\}$, where $P_n$ corresponds to the total number of points tracked in the $n^{\text{th}}$ object. For each $T_p^n$, we first identify the k-nearest neighbor tracks of the point as $T_k^n \in \{\text{k-NN}(T_p^n)\}$ from each frame. We find the distance to nearest neighbors, $d_{pk}^n$, as, $\left\| T_p^n - T_k^n \right\|_2$.

By computing scalar distances between neighboring points, the metric becomes invariant to global object motion and camera movement. Finally, to check the consistency or smoothness of neighbor point distances, we compute the moving average of $d_{pk}^n$ across frames and find the mean absolute error (MAE) of $d_{pk}^n$ with its moving average track $\bar{d}_{pk}^n$ as $\delta_{pk}^n$. This reveals the noisy trajectories in the neighboring tracks, capturing the distortions in each object. To obtain the object consistency error, we average $\delta_{pk}^n$ across the $k$ neighbours of each tracker and across all tracking points $P_n$. Finally, we take an average of object scores to obtain the final object inconsistency score.

## 5 EXPERIMENTS AND RESULTS

We evaluate the proposed metrics on the background scene consistency and foreground object consistency dimensions of our DynamicEval database. We evaluate our models on both the full dataset and a subset where all annotators agreed on their selection (full agreement).

**Video-level comparisons.** To compare video-level performance, we evaluate the methods on two metrics. Pairwise preference accuracy: The proportion of times a metric selects the same video as humans. Top-k video selection accuracy: The proportion of times the ground-truth best video is ranked within the top-k predictions of the metric. For Top-k evaluation, we obtain the ground-truth ranking of videos given a prompt through win-ratios (see supplementary for details). We compare our metrics with the baseline metrics proposed in VBench in Table 2. For the background consistency dimension of DynamicEval database, we evaluate VB-BG, VB-MS and our debiased motion smoothness metric (MS-Debias). The final pairwise scores are computed as an average of both the VB-BG and our MS-Debias to incorporate the best of both feature consistency and pixel-level consistency. Our final metric outperforms both baselines, highlighting the effectiveness of debiasing and multi-scale processing in video-level evaluation. For the foreground subject consistency dimension of DynamicEval database we evaluate VBench subject consistency (VB-SC) and our Track-FG metric. We apply the same logic as in background consistency to combine the strengths of feature- and tracker-level consistency. Our proposed metric outperforms the baseline, highlighting the advantage of explicitly focusing on the foreground objects and computing motion invariant metrics. Additionally, we qualitatively analyze the metrics (details in supplementary, Section D.2, and E.2.2).

**Model-level comparisons.** Similar to the model-level comparisons presented in VBench, we evaluate the ability of our metrics to rank the models. To obtain a model-level ground truth score, we compute the average of video win ratios across each model. To evaluate the correlation of model-level ground truth scores with metric scores, we compute their Pearson's Linear Correlation Coeffi-

Table 2: Performance Comparisons: The first table shows pairwise preference accuracy on the DynamicEval database, and the second table evaluates the Top-k video selection accuracy. On the full dataset, our method, MS-Debias outperforms the VB-BG by 2.2% points and on the full agreement subset by 2.4% points. Similar boosts can be seen for subject consistency pairwise accuracy.

| Background Consistency (Pairwise Acc.) | | |
| --- | --- | --- |
| Method | Full Dataset | Full Agreement |
| VB-BG | 56.0 | 59.3 |
| VB-MS | 53.7 | 56.8 |
| MS-Debias (Ours) | **58.2** | **62.7** |
| Subject Consistency (Pairwise Acc.) | | |
| VB-SC | 56.2 | 58.8 |
| Tracker-FG (Ours) | **58.2** | **62.7** |

| Background Consistency (Top-k) | | | | | |
| --- | --- | --- | --- | --- | --- |
| Method | Top-1 | Top-2 | Top-3 | Top-4 | Top-5 |
| VB-BG | 10.1 | 31.5 | 41.6 | 55.1 | 65.2 |
| MS-Debias (Ours) | **14.6** | **34.8** | **44.9** | **57.3** | **69.7** |
| Subject Consistency (Top-k) | | | | | |
| Method | Top-1 | Top-2 | Top-3 | Top-4 | Top-5 |
| VB-SC | 16.2 | 29.1 | 39.5 | 50.0 | 59.3 |
| Tracker-FG (Ours) | **19.7** | **31.4** | **41.9** | **55.8** | **62.8** |

Table 3: PLCC of model-level win ratios between the metrics and human preference. We evaluate the metrics on the full agreement subset.

| Method | Background Consistency | Method | Subject Consistency |
| --- | --- | --- | --- |
| VB-BG | 0.551 | VB-SC | 0.334 |
| MS-Debias (Ours) | **0.743** | Tracker-FG (Ours) | **0.772** |

cient (PLCC) in Table 3. Both MS-Debias and Tracker-FG outperform the baseline by a significant margin in terms of correlation with human judgments for selecting the best model.

Table 4: Performance of each metric when the pair contains different configurations of static-dynamic pairs.

| Background Consistency | | | Subject Consistency | | |
| --- | --- | --- | --- | --- | --- |
| Method | static-dynamic | dynamic-dynamic | Method | static-dynamic | dynamic-dynamic |
| VB-BG | 55.2 | 56.1 | VB-SC | 57.1 | 57.3 |
| MS-Debias | **58.0** | **56.7** | Tracker-FG | **60.3** | **58.1** |

**Pairwise preference on static and dynamic scenes.** Although our prompt suite is designed to generate dynamic scenes, a fraction of the generated videos are static, as shown in Figure 2b. We use this distinction to analyze the effectiveness of our metrics by partitioning the dataset into static and dynamic videos. In a pairwise comparison, there are three scenarios: both videos are static (static-static), one of them is static (static-dynamic), and both are dynamic (dynamic-dynamic). As the occurrence of static-static pairs is very low, we omit evaluation on this subset. We evaluate the preference performance on each of these subsets in Table 4. The experimental results show that our proposed metrics consistently outperform the baselines when dynamic scenes are present.

# 6 DISCUSSIONS AND LIMITATIONS

The practical use of our proposed metrics relies on off-the-shelf vision models for optical flow, point tracking, and segmentation. Large-scale pre-training enables these vision models to capture visual primitives (edges, textures, objects, semantics) that transfer across domains. Notably, RAFT and CoTracker, despite being trained on synthetic 3D scenes, generalize well to out-of-distribution real videos, and naturally to generated videos as well. Prior state-of-the-art works in evaluation metrics for T2V (Huang et al., 2024a; Liu et al., 2024b; Huang et al., 2024b; Asim et al., 2025) further show that such models remain effective for evaluating generated videos. When generated videos contain distortions, these models exhibit larger errors and help identify distorted videos, thus yield significant inconsistency scores, as expected. Thus, off-the-shelf models are crucial in zero-shot evaluation of generated videos. There are some limitations in our work, namely that our metrics rely heavily on the correct estimates of the optical flow and point tracks that may not always be the case for generated videos. The optical flow computation and the CoTracker are trained on real videos while in our work they are to be computed on the generated videos which can lead to erroneous computation. However, we note that given the zero-shot and plug-and-play nature of our metric computation, the external models are replaceable components and as their reliability and quality improve, the robustness of our metrics also will improve.

## ETHICS STATEMENT

This work complies with the ICLR Code of Ethics. Our research does not involve human subjects, personal data, or sensitive content. The methods we propose could potentially be used in real-world applications in evaluating video models. We do not foresee immediate dual-use risks or malicious applications associated with our work.

## REPRODUCIBILITY STATEMENT

We are committed to ensuring the reproducibility of our results. To facilitate this:

- We intend to release the source code required to reproduce the results in the paper.
- All datasets used or created in our experiments will be made available publicly in due course of time.
- We will release all hyperparameters used to develop the metrics specified in section 4.1 and 4.2.
- Experiments were conducted on a single NVIDIA A100 GPU.

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
