# OpenReview forum: "DynamicEval: Rethinking Evaluation for Dynamic Text-to-Video Synthesis"
_ICLR.cc/2026/Conference — ICLR 2026 Conference Withdrawn Submission_

### Official Review · Reviewer_oNBz · 2025-10-29

**Soundness:** 2
**Presentation:** 2
**Contribution:** 2
**Rating:** 2
**Confidence:** 5

**Summary:**

The paper proposes DynamicEval, a benchmark consisting of systematically curated prompts emphasizing dynamic camera motion.

**Strengths:**

1. The paper considers evaluating video generation from a different aspects, i.e., dynamic.

2. The paper is easy to follow.

3. The paper fulfills 9 pages.

**Weaknesses:**

1. There is already some works focusing on dynamic aspects: Evaluation of Text-to-Video Generation Models: A Dynamics Perspective (NeurIPS 2024). The paper either cites the paper or discuss the differences with that. It is unacceptable becausue they are too similar.

2. The writing is poor. As shown in Fig. 1 and 2, the fonts in figures are too small to be checked.

3. The evaluation is not comprehensive. Compared to Vbench, which evaluates video generation from many aspects. The paper only considers the dynamic aspect.

4. The prompts in the dataset are not from the real world datasets, such as VidProM. It is sythesized which limits the practical use of this paper.

5. There is no ethnic approvement for using human sources to evaluate. Does they work fairly? Is there any forced labor happens?

**Questions:**

NA

---

### Official Review · Reviewer_V5mv · 2025-10-30

**Soundness:** 3
**Presentation:** 3
**Contribution:** 2
**Rating:** 4
**Confidence:** 2

**Summary:**

This paper introduces ​**DynamicEval**​, a new benchmark and metric suite for evaluating *text-to-video (T2V)* models under ​**dynamic camera motion**​, a setting largely ignored by existing benchmarks such as VBench and EvalCrafter.

DynamicEval contributes:

1. **A dataset** of 3 000 videos from 10 T2V models, with 45 000 pairwise human annotations on two key dimensions — **background (BG) consistency** and **foreground (FG) object consistency** — focusing on scenes with explicit camera motion.
2. ​**Two fine-grained automatic metrics**​:
   * ​**MS-Debias**​, a debiased optical-flow–based background consistency score that masks foregrounds and occlusion areas in VBench’s Motion Smoothness metric.
   * ​**Tracker-FG**​, a CoTracker-based object-consistency score measuring local deformation smoothness among point tracks within each object.
3. **Empirical results** showing higher human correlation (≈ +2 pp video-level accuracy, +0.2 PLCC) than prior metrics, and a curated prompt suite emphasizing diverse camera motions.

The work aims to improve *video-level* evaluation and offers a resource for developing future T2V metrics.

**Strengths:**

#### 1. Originality

* Tackles an underexplored aspect — **dynamic camera motion evaluation** — that is critical for cinematic T2V quality.
* Introduces a new **procedural prompt-generation pipeline** and a **large-scale human study** focused on motion-rich prompts.
* The **debiasing of optical-flow error maps** and the **track-based FG metric** are creative, interpretable extensions of existing metrics.

#### 2. Clarity

* Writing is well structured and readable; figures (e.g., error-map visualizations) illustrate ideas effectively.
* Equations and algorithmic steps for both metrics are explicit and reproducible.

#### 3. Significance

* Provides a **missing benchmark component** for evaluating dynamic video generation—a growing focus in commercial and open research T2V models.
* The metrics are **plug-and-play** and interpretable, likely to become standard diagnostic tools.
* Potentially useful for *reinforcement-learning-from-video-feedback* and model-selection pipelines.

#### 4. Quality

* Methodology is rigorous: ablation across dynamic/static subsets, full-agreement subsets, and multiple baselines.
* Clear quantitative improvements support claims.
* Dataset construction and reliability checks follow perceptual-quality-assessment best practices.

**Weaknesses:**

* **Limited Scope of Evaluation Dimensions**
  Only background and foreground consistency are addressed. Other key axes (temporal coherence of lighting, motion realism, text-prompt faithfulness) are untouched; the benchmark might thus be incomplete as a holistic evaluation suite.
* **Reliance on Pre-trained Vision Modules**
  Although acknowledged, heavy dependence on RAFT and CoTracker trained on real videos may bias results when synthetic artifacts violate their priors. Experiments quantifying metric stability under flow/track noise would strengthen robustness claims.
* **Dataset Bias and Size**

  * 100 prompts × 10 models × 3 videos yields 3 000 clips ≈ short scenes — perhaps insufficient diversity for generalization.
  * Prompts are GPT-generated; linguistic diversity and style variance may still be limited.
* **Statistical Analysis**
  Improvements of 2 pp are meaningful but modest; confidence intervals or significance testing (e.g., bootstrap CI of human-metric correlation) are missing.
* **Accessibility / Reproducibility Timing**
  The dataset and code are “to be released,” which weakens verifiability at review time.

**Questions:**

**Broader Impact** – Could DynamicEval incentivize models to over-smooth motion to maximize scores? Any mitigation?

---

### Official Review · Reviewer_TTWy · 2025-11-01

**Soundness:** 2
**Presentation:** 1
**Contribution:** 1
**Rating:** 2
**Confidence:** 5

**Summary:**

This paper propose DynamicEval, a new benchmark comprising 100 dynamic-focused prompts, 3,000 generated videos from 10 models, and 45,000 human pair-wise preference annotations. Concurrently, it proposes two new metrics: MS-Debias for background (BG) consistency, which "debiases" VBench's motion smoothness by masking out foreground objects and occlusion artifacts , and Track-FG for foreground (FG) consistency, which uses a point tracker (CoTracker) to analyze the smoothness of relative distances between internal object points. Experiments show these new metrics achieve over 2% points higher correlation with human preferences than baselines.

**Strengths:**

The primary strength of this work lies in its improvement of the metrics for background consistency and foreground (subject) consistency.

**Weaknesses:**

1. Limited Evaluation Scope: The paper reduces the complex, multi-dimensional problem of "dynamic video quality" to only two dimensions: FG and BG consistency. This is an overly narrow view that ignores other critical aspects of video generation, such as: Text-Video Alignment, Motion Plausibility, Aesthetic Quality, etc.
2. A Flawed "Rigid-Body" Assumption: The core design of the Track-FG metric is fundamentally flawed. By assessing consistency based on the "smoothness of distances" between tracked points, the metric implicitly assumes that foreground objects are rigid or near-rigid. This is factually incorrect for most real-world scenarios. Non-rigid objects like a person talking (facial deformation), a flag waving, water rippling, or cloth moving in the wind are all "consistent" and high-quality, yet the distances between points on their surfaces should change dramatically. The Track-FG metric would incorrectly penalize these correct, high-quality generations as "inconsistent" or "distorted.
3. Methodological Loophole: "Cheating" via Static Videos: The paper's goal is to evaluate "dynamic camera motion". However, the authors admit in their own analysis (Fig. 2b) that many models—especially open-source and older ones—failed to follow the dynamic prompts and instead generated static videos. The authors' decision to "retain static videos in the evaluation"  creates a critical methodological flaw. A static video will naturally achieve a near-perfect score on any "consistency" metric. This means models are effectively rewarded with higher consistency scores for failing to adhere to the benchmark's primary goal (dynamic motion). This "static video loophole" invalidates the benchmark's claim of evaluating dynamic scenes.
4. Incremental Novelty: The paper's contributions are largely incremental. The work consists of "debiasing" an existing metric (VB-MS) and replacing another (VB-SC) with a different technique (tracking) to measure the same dimension of consistency. The paper does not, as its title "Rethinking Evaluation" suggests, propose any new, fundamentally different evaluation paradigms or dimensions.
5. Error Propagation Risk: Both metrics rely on a complex and fragile "toolchain" of off-the-shelf models (RAFT , CoTracker , LLMs, GroundingDINO , and SAM-2 ). As the authors partially acknowledge, these tools were trained on real videos. Their reliability on generated videos, which may contain unique artifacts, is questionable. This creates an ambiguity: a low score could mean the video is poor, or it could mean the evaluation tool (e.g., CoTracker) failed to track points on a distorted video.

**Questions:**

see weakness above

---

### Official Review · Reviewer_ezqH · 2025-11-01

**Soundness:** 3
**Presentation:** 2
**Contribution:** 2
**Rating:** 4
**Confidence:** 3

**Summary:**

This paper proposes DynamicEval, a benchmark focused on evaluating dynamic motion, which includes 100 prompts with camera motion descriptions. To assess motion quality, two new metrics are introduced: Background Consistency and Subject Consistency. Specifically, pixel-level tracking is used to capture more fine-grained spatial awareness and long-term temporal consistency. Additionally, the paper determines whether a video is static or dynamic based on pixel-level point movement. The proposed metrics were tested for alignment with human annotations, and experiments show that they achieve a 2% improvement in agreement with human judgments compared to baseline metrics.

**Strengths:**

1. This paper proposes a benchmark for evaluating video motion quality, which includes 100 prompts describing camera motion.

2. The paper uses point tracking to capture finer spatial perception and long-term temporal consistency at the pixel level.

3. The paper is clearly and logically written, outlining the research gap and explicitly defining the evaluation metrics.

**Weaknesses:**

1.Compared to VBench, the measurement of foreground and background consistency in this paper replaces RAFT with CoTracker and adds masking. The construction of the metrics in this paper is incremental relative to VBench.

2.The evaluation of camera motion in this paper is simply divided into static and dynamic. In video generation, camera motion involves complex 3D spatial relationships, so this simple classification makes the evaluation metrics rather limited.

3.Limited evaluation dimensions: As a benchmark paper, the evaluation primarily covers only three aspects—foreground consistency, background consistency, and whether the scene is dynamic.

4.The evaluation method relies on multiple external models, such as detection, segmentation, and tracking, making it susceptible to the limitations of these models. If one or more of these external models fail, the proposed metrics can be easily affected.

**Questions:**

Please refer to the Weaknesses section.

---

### Note · Authors · 2025-11-13

I have read and agree with the venue's withdrawal policy on behalf of myself and my co-authors.